# Evaluation of the Effects of Particle Sizes of Silver Nanoparticles on Various Biological Systems

**DOI:** 10.3390/ijms21228465

**Published:** 2020-11-11

**Authors:** In Chul Kong, Kyung-Seok Ko, Dong-Chan Koh

**Affiliations:** 1Department of Environmental Engineering, Yeungnam University, Gyungsan 38541, Korea; ickong@ynu.ac.kr; 2Geologic Environment Division, Korea Institute of Geoscience & Mineral Resources (KIGAM), Daejeon 34132, Korea; chankoh@kigam.re.kr

**Keywords:** bioassays, particle size, silver nanoparticles, toxicity

## Abstract

Seven biological methods were adopted (three bacterial activities of bioluminescence, enzyme, enzyme biosynthetic, algal growth, seed germination, and root and shoot growth) to compare the toxic effects of two different sizes of silver nanoparticles (AgNPs). AgNPs showed a different sensitivity in each bioassay. Overall, the order of inhibitory effects was roughly observed as follows; bacterial bioluminescence activity ≈ root growth > biosynthetic activity of enzymes ≈ algal growth > seed germination ≈ enzymatic activity > shoot growth. For all bacterial activities (bioluminescence, enzyme, and enzyme biosynthesis), the small AgNPs showed statistically significantly higher toxicity than the large ones (*p* < 0.0036), while no significant differences were observed among other biological activities. The overall effects on the biological activities (except shoot growth) of the small AgNPs were shown to have about 4.3 times lower EC_50_ (high toxicity) value than the large AgNPs. These results also indicated that the bacterial bioluminescence activity appeared to be an appropriate method among the tested ones in terms of both sensitivity and the discernment of particle sizes of AgNPs.

## 1. Introduction

With the development of nanotechnology-based products in many areas of the industrial market, the variety of types of nanoparticles (NPs) in commercial products has led to their increased presence in the environment [1,2]. Among the increasing number of products and types of NPs, silver-based NPs have been widely used in various industrial fields due to the growing capacity for synthesis and manipulation across diverse fields, such as in daily products and medical products, electronics, foods, and industrial purposes, because of their unique physical and chemical properties [3,4,5,6]. AgNPs include any silver-containing particles with enhanced activity due to their nanosize features, such as clusters of Ag^0^ and metallic silver atoms, manufactured for practical usages [7]. Yan and Chen [8] reported that approximately 25% of all nanotechnology consumer products contain AgNPs. Products, such as socks, textiles, baby products, and washing machines, are also receiving considerable attractions. AgNPs are detected widely in various environments at a high level of accumulation, as well as in the effluent of domestic wastewater [9,10]. In addition, soluble silver ions (Ag^+^) can be leached from AgNPs, and highly toxic silver ions are bioaccumulative and persistent to organisms in the aquatic environment [8,11]. Therefore, great concerns about their safety and environmental toxicity have arisen due to the potential adverse effects of AgNPs on health and ecosystems [9,12,13,14,15].

The toxicity of NPs is generally influenced by their physicochemical properties, such as their particle size, surface characteristics, shape, coatings, capping, solubility, and reactivity [16,17,18,19]. Various toxicities of AgNPs have been observed according to the particle size and particle shape [20,21,22]. Effects of surface coating on toxicity have also been reported due to their influence on the dissolution of AgNP [23]. Capped AgNPs with starch and bovine serum albumin showed deleterious effects in fish embryos compared to the exposure of uncapped AgNPs [24]. Small silver NPs produce higher reactivity and show higher genotoxicity [25]. Monikh et al. [26] observed that the combined effect (size, shape, and ecocorona) also controls the attachment and physical toxicity to organisms. Although the toxic mechanisms of NPs are not precisely understood or many contradictory findings have been reported, the commonly proposed mechanisms involve either solubilized metal ions in the solution or direct uptake by cells, followed by DNA damage, the deactivation of cellular enzymes, the disruption of cell membrane, and altered cellular redox balance due to the production of reactive oxygen species (ROS) [6,7,27,28,29].

Research data on the toxicity of NPs have reported different or opposite results according to the biological systems adopted [3,30]. For the toxicity evaluation of AgNPs, organisms such as anthropods, plants, earthworms, and enzymes have been observed [31,32,33,34,35,36]. To date, various organisms at different trophic levels and their specific metabolic processes have been tested to evaluate the toxicity of NPs [29,37,38,39], for example, toxic effects on the growth of algae and crustaceans for the widely used TiO_2_ and ZnO NPs were exhibited, while opposite results (not toxic) to the activity of *Vibrio*, *Daphnia*, or *Thamnocephalus* species were shown [40,41,42]. The complexity of biological systems may also be related to the toxicity of NPs [43,44,45,46,47]. Therefore, comparative analysis using different levels of biological systems are recommended to investigate the toxic effects of NPs. The evaluation of the plant processes (e.g., biomass, germination, and root and shoot growth) is considered to involve short-term assays as there is a rapid response to acute toxicity in environmental biomonitoring, which is particularly relevant when contaminants are present in soil [48]. For example, significant changes of seed germination, biomass, and leaf surface area, which are common parameters for assessing the phytotoxicity of AgNPs in plants, were observed after exposure to AgNPs [8,49]. Thuesombat et al. [2] also reported that the levels of rice germination and root and shoot growth were both decreased with increasing concentrations, as well as a higher uptake when treated with smaller AgNPs. The primary producer that plays an important role in the environment, microalgae, is also commonly used to examine the effects of NPs when contaminants are exposed in the aquatic environment [50]. AgNP showed a negative effect on freshwater and marine green algae [50]. In this case, the formation of different agglomerates, leading to physical and chemical alterations, may be the possible reason for the different sensitivities between tested algal species. Quite a number of studies have also been conducted to determine the antibacterial activity of AgNPs [51]. Due to the assay-dependent effects of NPs, an understanding of each test organism’s toxic mechanism and sensitivity as well as the choice of suitable bioassay organisms and metabolic processes is important to properly assess toxicity. In addition, the combined results of several bioassays can provide information for the more accurate bioassessment of NPs rather than those of a single bioassay. As the fate and effect of NPs may also vary depending on the physicochemical properties of the NPs and experimental conditions, the effect of these properties on the NPs will also need to be carefully controlled for the bioassessment of NPs in a laboratory test system.

The main purpose of this investigation was to evaluate the toxic effects of two different particle sizes of AgNPs using seven biological methods: bacterial bioluminescence activity, enzyme (β–galactosidase) activity, enzyme biosynthetic activity, seed germination, algal growth, and root and shoot growth. After analyzing the test results, a suitable method for the toxicity of Ag-NPs in terms of both sensitivity and the discernment of particle sizes was also suggested.

## 2. Results

### 2.1. Effect of AgNPs on Three Bacterial Activities

The effects of two particle sizes of metallic AgNPs were examined based on the three bacterial activities of bioluminescence, enzyme, and enzyme biosynthesis. Based on the preliminary tests, the dose-response curves of AgNPs for bioluminescence activity were determined in the range of 0–100 mg/L for both particles (Figure 1). The control produced a mean of 1490–1611 RLU of bioluminescence at the beginning of the incubation, depending on the experimental sets. To compare the changes of bioluminescence activity over time between two particle sizes, the representative results of the percentage of relative activity at 50 mg/L are shown in Figure 1a. A large decrease in activity was observed during the initial 30 min exposure period, showing 24% and 51% of control activity for size#1 and size#2 particles, respectively. After 2 h of exposure, nearly complete inhibition was observed with the exposure of size#1, while 22% activity was observed with size#2 at 50 mg/L (Figure 1a). However, similar changes of bioluminescence activity appeared at 100 mg/L for both particles, showings a high inhibitory effect on bioluminescence activity (less than 20% relative activity) (Figure 1b). The mean bioluminescence activity at 1 h and 1.5 h of exposure was used for the comparisons of the effect for two particle sizes at different concentrations (Figure 1b). For example, the activities of size#1 AgNPs decreased from 20 ± 0.6% and 11 ± 0.6% of the controls at 50 and 100 mg/L, respectively, whereas exposure to size#2 AgNPs resulted in a relative activity of 107 ± 9.8% and 19 ± 1.3% of the controls at 50 and 100 mg/L, respectively (Figure 1b). No significant reduction in bioluminescence activity was appeared at the conditions up to 50 mg/L of size#2, while a significant reduction was exhibited for size#1, showing 20 ± 0.6% activity at 50 mg/L of size#1 AgNPs. The EC_50_ calculated on the basis of the mean bioluminescence activity showed values of 10 (7.72–13.75) mg/L and 136 (124.4–147.6) mg/L for the particle size#1 and size#2, respectively, showing about 12-fold greater toxicity with particle size#1 than particle size#2 (*p* = 0.0001) (Table 1).

The effects of AgNPs on the enzymatic and biosynthetic activity of β-galactosidase were determined at up to 1000 mg/L of AgNPs (Figure 2). Both activities were determined by the color changes (yellow to redish) of chromogen in solution. The mean absorbance of the control (no AgNP exposure) for both activities was in the range of 1.68 to 1.79, and no stimulations were observed under all tested conditions. Enzymatic activity with size#1 decreased sharply from 92% to 28%, while a gentle decrease from 89% to 77% in activity was observed with size#2 at exposure concentrations of 50 and 200 mg/L (Figure 2a). In contrast to the enzymatic activity, the effects on the biosynthetic activity showed different patterns, changing from 64% to 29% and from 59% to 55% of the activity of the control for size#1 and size#2 at 50 and 200 mg/L, respectively (Figure 2b). At 1000 mg/L, the enzymatic activity appeared to be 14 ± 1.1% (86% toxicity) and 58 ± 2.1% (42% toxicity) for size#1 and size#2, respectively, showing about twofold higher toxicity for size#1 compared to size#2, while a greater effect on the biosynthetic activity compared to the enzymatic activity was observed, showing 7 ± 2.6% and 25 ± 1.3% of the control activity for size#1 and size#2, respectively. The EC_50_ values calculated for the enzymatic activity were 148 (125–176) mg/L and 1170 (930–1472) mg/L, while the values for the biosynthetic activity were 85 (70.2–99.8) mg/L and 131 (110–152) mg/L for size#1 (*p* = 0.0029) and size#2 (*p* = 0.0036), respectively (Table 1). The small particle sizes showed about 8- and 1.5-fold greater toxicity than the larger particles for the enzymatic and biosynthetic activity, respectively (Table 1).

### 2.2. Effect of AgNPs on Algal Growth

The effects of different sizes of AgNPs were evaluated at up to 100 and 200 mg/L of AgNP by the activity of three endpoints (absorbance, cell count, and chlorophyll content) for size#1 and size#2 particles, respectively. The average absorbance, cell count, and chlorophyll content of algal cultures of the control (no AgNP exposure) were shown as 1.9 × 10^6^/mL, 1.17, and 5.95 g/m^3^, respectively. Correlation coefficients (R^2^) between two endpoints were observed as 0.723 (cell count/chlorophyll content), 0.788 (absorbance/cell count), and 0.876 (chlorophyll content/absorbance) (Figure 3). No stimulation of algal growth was observed under the conditions tested. Under the tested conditions for both particle sizes, relative activities of cell count, absorbance, and chlorophyll contents ranged from 12 to 98%, 22 to 88%, and 5 to 98% compared to the control, respectively (Table 2). A rapid decrease of activity was observed up to 50 mg/L of AgNPs, following slight changes of activity for both particle sizes. For example, the average activity for both particles was about 21% (toxicity 79%) at 100 mg/L AgNPs (Table 2). The EC_50s_ calculated were 48 (42.2–53.8) mg/L and 60 (46.0–79.9) mg/L for size#1 and size#2, respectively. The small particle sizes (size#1) showed an about 1.3-fold higher inhibitory effect on algal growth than the large particle sizes (size#2) (*p* = 0.2778) (Table 2).

### 2.3. Effect of AgNPs on Seed Germination and Root and Shoot Growth

Based on the pre-range tests, the effects of the two particle sizes on seed germination and root and shoot growth were evaluated at up to 200 mg/L of AgNPs. A slight stimulation of germination activity was observed at lower concentrations of AgNPs (20 mg/L) for both particle sizes. The activity of seed germination for both particle sizes showed a similar tendency with increasing exposure concentration, showing activity in the range of 3 to 117% of the exposure ranges (Table 2). There was also no observable difference of EC_50_ values between the two particles, showing values of 138 (130.5–145.6) mg/L and 137 (133.4–141.4) mg/L for size#1 and size#2, respectively (*p* = 0.8492) (Table 2).

For the control of other response endpoints of root and shoot growth, an average of 75 ± 18.1 mm and 21 ± 3.1 mm growth was observed for root and shoot after 4 days of incubation, respectively. In the experimental groups under the tested conditions, the RSL and RRL ranged from 71 to 100% and 26 to 43% of the control activity, respectively. No considerable inhibition and differences of shoot growth were observed for both particle sizes under the tested concentration ranges, showing above 71% of the activity of the control under all conditions. However, a sharp increase of inhibition of the root growth was observed for both particle sizes up to 20 mg/L of AgNPs exposure, showing an RRL of 28 ± 4.2% and 43 ± 16.1% for size#1 and size#2 particles, respectively (Table 2). At concentrations higher than 20 mg/L, no further appreciable inhibition in root growth was observed up to the maximum exposure concentration of 200 mg/L, indicating 26–37% of RRL for both particle sizes. The calculated EC_50_ values for root growth were 10 (7.4–13.5) mg/L and 13.4 (8.85–20.14) mg/L for size#1 and size#2 particles, respectively, showing about 1.3-fold higher root growth toxicity for size#1 (*p* = 0.4080) (Table 2).

## 3. Discussion

The effect of two different particle sizes of AgNP was evaluated based on seven biological activities. Researchers reported that several characteristics of NPs (shape, particle size, NP types, solubility, crystallinity, chemical composition, chemistry, and impurities) and environmental factors affect the toxicity of NPs [7,52,53,54]. The toxicity of NPs may change for organisms within the same taxonomical group (i.e., bacteria of different species) and very different groups of organisms (i.e., bacteria vs. human cells), as well as based on the composition of the culturing medium for test organisms. Various bacterial activities have been used to assess the toxic effects of environmental contaminants due to their simplicity of handling, reasonable sensitivity, and reproducibility. In this investigation, three bacterial activities (bioluminescence, enzymatic activity, and the biosynthesis of enzyme) were adopted to evaluate the effects of the particle size of the AgNPs. Although the effects varied depending on the concentrations and bacterial activities tested, small AgNPs showed statistically significantly higher toxicity than large AgNPs for all bacterial systems tested, showing 1.5- to 13-fold higher inhibitory effects than large particles on the basis of the calculated EC_50s_ (*p* < 0.0362) (Table 1 and Figure 4). These observed different effects may be related to the particle size characteristic of NPs affecting the intimate contact between cells and NPs [41]. For example, Cho et al. [4] reported that the toxicity of 60 and 100 nm AgNPs to eukaryotic cells was weaker than that of 10 nm AgNP. In addition, solubilized metal ions from AgNPs may be an important factor for the toxic effects on the bacterial activity; in particular, cationic metal ions from NPs prefer to interact with the negative charges of Gram-negative bacterial cell walls, potentially leading to an increased negative effect on enzyme biosynthesis or the DNA damage of cells [55,56]. An increase in the permeability of bacterial cell membranes also accelerated the penetration of AgNPs into the cytoplasm and disturbances in cellular functions [57]. In this study, soluble silver concentrations were measured to determine the degree of the effect of the soluble silver metal on the bacterial activity. Very low and wide ranges of dissolved silver concentrations were detected after the incubation of experiments, ranging from 2 (set of bacterial bioluminescence at 10 mg/L) to 114 μg/L (set of enzymatic activity at 1000 mg/L), which correspond to a minimum of 0.011% to a maximum of 0.092% of the initial amended concentration (average 0.037 ± 0.0284%). Experimental sets with different particle sizes showed no considerable differences of the dissolved concentration. These data suggest that the contribution of the soluble silver ions on bacterial activity could be very low or insignificant under tested conditions. Researchers have also reported that the effect of metal ions of NPs does not account for the total toxicity of NPs [58]. Therefore, the toxicity of AgNPs is influenced not by soluble metal ions but mostly by the particle properties; however, the soluble metal concentration of the NPs could vary by the type of NPs, the environment, or the medium in which the NPs are exposed. Therefore, the toxicity of NPs may be induced both by the released metal cations and by other physical and chemical characteristics of NPs. The relative contributions for each factor may behave differently depending on the characteristics of NPs, as well as the experimental sets and exposed environmental conditions [59,60].

The growth of algal species is commonly used to examine the effect of contaminants on environmental systems (e.g., the reduction of chlorophyll—a content or pigment-protein complexes of algal growth) [61]. In this study, the effects of AgNPs were evaluated by the average of three endpoints (absorbance, cell count, and chlorophyll content) of algal growth. The observed high correlation coefficients (R^2^) between all two endpoints (0.723–0.876) suggested the possible usage of just one measured endpoint for the assessment of AgNPs. No statistically significant differences in toxicity were observed between both particles exposed to the algal culture under tested conditions. Although no statistically significant differences were considered between the two EC_50_ values calculated, a small particle size showed an about 1.3 times higher inhibitory effect on algal growth than a large particle size (*p* = 0.2778). A previous study in our laboratory with Co-NPs also showed an approximately twofold higher inhibition with small particles (10–30 nm) compared to larger ones (50–80 nm) on algal growth. Researchers have reported that the effects on algal growth are related with many factors of NPs, such as the solubilized metal ions, the particle sizes, the reactivity to photosynthetic enzymes, chemical composition, and the adhesion, etc. Some of these factors may lead to the induction of ROS, the inhibition of biomass, the partial inhibition of photosynthetic activity, an increase in cell size, cell membrane disruption, a reduction in nutrient uptake, and the generation of photocatalytical reactive species [42,50,53,62]. The cell walls could be a site for interactions and a barrier for the transport of NPs into algal cells, influenced by the structure, the particle sizes, or type of NPs [63]. Navarro et al. [61] reported that only particles smaller than 20 nm are likely to enter the cytoplasm after passing through the cell membrane due to the ability of algal cells to be sieved out of the NPs. In this investigation, silver concentrations in the solutions of a few experimental sets were analyzed to understand the contribution of the soluble metal ions of each particle size on the algal activity. As with the bacterial experimental sets, a low concentration of dissolved silver (0.2–0.5 mg/L corresponding to 0.1% to 0.5% of 20 to 100 mg/L Ag-NPs tested) was detected and no observable differences were observed between the two particle sizes, suggesting partial or insignificant contributions to the algal toxicity. Therefore, it can be speculated that the toxicity of different particle sizes of AgNPs is mainly induced by the physiochemical properties of NPs [58]. Zhang et al. [64] reported that the toxicity of AgNPs largely occurred due to the joint effect of particle form and released silver ions. However, the relative contributions between the particles and the release or availability of solubilized metal ions, which also can be affected with media component, has not yet been well described and may behave differently depending on the tested conditions [59,60]. Other possible causes for the toxicity differences of particle sizes in algae cells include the formation of large aggregates due to the binding to the cell surface or the interaction with cell wall, which could inhibit cell division [50,65]. NPs adsorbed on the cell wall, which may be influenced by the particle sizes, were able to lead to the formation of agglomerates, and these may have participated in the toxicological effects [50,66]. Pikula et al. [67] also suggested that the toxicity of NPs on different algal species depends on the interaction of cell wall components.

The growth potential of seed germination and root and shoot growth are also commonly adopted for the assessment of pollutants [3,8]. In this investigation, the activity of root growth was the most sensitive to AgNPs, followed by germination and shoot growth, possibly due to their contact with and retention by the root tissues. No statistically significant differences were observed between the effects of the two particle sizes (Figure 4). Although many studies have reported that small-sized AgNPs cause higher toxicity than large-sized AgNPs, this correlation between toxicity and AgNP size is not always true for all AgNP exposure conditions [2]. The low toxicity (high EC_50_; Table 1) of seed germination compared to all other activities tested in this study is probably due to the matter of penetration. NP particle penetration via the seed coat to the embryonic tissue may be inhibited by the effect of enzymes for germination as well as other activities, which are important influencing factors on the effect of seed germination. A lack of inhibitory effects on shoot growth was observed due to the absence of contact with NP particles, suggesting that shoot growth is an inappropriate method for the assessment of NPs. Among the seven biological activities tested in this study, the most significant effects were observed for root growth. Researchers have also reported that small-sized AgNP particles had greater effects on root growth and browning, higher accumulation in tissues, and increased susceptibility [68,69]. In this study, dissolved silver metal ions were also determined to examine the influence of soluble ions on the toxicity of root growth. Very low soluble silver metal concentrations, ranging from 2 to 60 μg/L exposed to 10 to 50 mg/L AgNP (corresponding to less than 0.2% of exposed AgNPs), were observed in the experimental solution of root growth, and no observable differences were observed between sets with different particle sizes, indicating an only partial or, more likely, insignificant contribution toward root toxicity. Papis et al. [70] also reported that the soluble ion concentrations of NPs in the culture medium were very low and insignificant concentrations. Lee et al. [71] reported that the toxicities of bean and wheat were attributable to the NPs (1000 mg/L Cu-NPs) rather than the solubilized Cu ions (0.3 mg Cu ions/L). However, the NPs were able to dissolve more efficiently once inside the cells, acting as an ion reservoir that in turn could cause toxicity by increasing ROS production, the oxidization of proteins, and causing oxidative DNA damage, which could vary according to the cellular type, biological system, and experimental conditions (dose, exposure length, etc.) [70]. Some studies reported that NP-coupled proteins can bind to cell surface receptors, enhance cellular uptake, and trigger intracellular signaling pathways; otherwise, a “protein corona” effect could significantly reduce the interactions with cells or stabilize NPs against dissolution [29,37,72]. Some potential contaminants of engineered NPs (e.g., synthesis, breakdown, and surface functionalization), culture media, and growth conditions may also cause toxicity unrelated to their nanoscale dimensions [55,73].

## 4. Materials and Methods

### 4.1. Chemicals, Preparation, and Analysis

The two different particle sizes of AgNPs (20–40 nm and 80–100 nm, further named as size#1 and size#2, respectively) used in this study were obtained from US. Research Nanomaterials, Inc. (Houston, TX, USA). The Ag-NPs were solubilized directly in distilled water (pH 7.8) and dispersed by sonication for 10 min prior to use. All other reagent grade chemicals were purchased from Sigma Chemical Co. or Aldrich Chemical Co. (St. Louis, MO, USA). The analyses of Ag ions were performed using a method similar to the study of Rahman et al. (2018). The AgNP solution samples were filtered (Advantec MFS Inc., Dublin, CA, USA) and treated with 5% HNO_3_ to prevent precipitation, and then the solubilized Ag ions in the experimental sets were determined [74]. The concentration of silver in the AgNP solution was measured with an inductively coupled plasma optical emission spectrometer (ICP-OES Perkin-Elmer Optima 7300DV, Waltham, MA, USA). The operating conditions were as follows; a wavelength of 328.068nm, a plasma gas flow rate of 15 L min^−1^, a RF generator power of 1.65 kW, a nebulizer gas flow rate of 0.65 L min^−1^, an auxiliary gas flow rate of 0.2 L min^−1^, and a solution uptake rate of 1.5 L min^−1^.

Each of the experimental results was compared to its corresponding control. The Student’s *t*-test (http://www.graphpad.com) was used for the statistical analysis for the experimental groups. When the probability of the result assuming the null hypothesis (*p*) was less than 0.05, the statistical significance was accepted.

### 4.2. Effect of AgNPs on Bioluminescence Activity

The *E. coli* DH5α RB1436 mutant strain, producing bioluminescence during growth due to a plasmid with a constitutive promoter, was used. The preparation of this strain for the toxicity studies was as described in previous studies [75]. For the bioassay, AgNP solution (9 mL) was combined with the diluted bacterial suspension (1 mL), and bioluminescence activity (relative light units (RLU)) was measured every 30 min during incubation periods. The toxicity was evaluated in relation to the average bioluminescence intensity, measured at 1 h and 1.5 h. Bioluminescence intensity was measured with a Turner 20/20 luminometer (Turner Design Inc., Sunnyvale, CA, USA; maximum detection limit of 9999 RLU).

### 4.3. Effect of AgNPs on Enzymatic and Biosynthetic Activity of β-galactosidase

The *Escherichia coli* EW1b mutant strain was used for the production of the enzyme (β-galactosidase). The strain cultured overnight in lactose broth (LB) medium (100 rpm, 35 °C) was further diluted 20-fold in LB medium and then cultured until an optical density (OD_550_) of 0.7. The culture was left for 4 h for enzyme induction with Isopropyl-β-D-thiogalacto-pyranoside (IPTG; stock solution 1000 mg/L; final concentration 100 mg/L). The enzyme-induced culture was centrifuged twice, and the pellet was washed with a sterilized 0.85% NaCl solution. After the removal of the supernatant, the pellet was resuspended with 1% sterilized trehalose, distributed into vials and stored in a deep-freezer. The enzymatic activity of β-galactosidase was determined by the color changes of an added chromogen (chlorophenolred-β-D-galactopyranoside (CPRG)) using a 96-well microplate reader. The procedure for the evaluation of the effect of AgNPs on enzyme biosynthesis was adopted based on the results of several preliminary tests. Detailed test protocols for the enzymatic and biosynthetic activity of enzyme were described in previous studies [75].

### 4.4. Effect of AgNPs on the Algal Growth Activity

The effect of AgNPs was evaluated based on the algal growth (*Chlorella vulgaris*), which was obtained from the Korean Culture and Tissue Collection (KCTC AG10002). After 3 days of incubation, the growth inhibition was assessed by measuring three endpoints (cell counts, absorbance, and chlorophyll content). The preparation of the algal culture and test protocol for this assay was as described in previous studies [75,76].

### 4.5. Effect of AgNPs on Seed Germination, Root and Shoot Growth

*Lactuca sativa* L. was purchased from a commercial seed company (Nongwoo Bio., Suwon, Korea). Twenty seeds were surface-sterilized in 3% H_2_O_2_ for 10 min and rinsed with distilled water, and these were placed on the filter paper in a Petri dish with 5 mL of solution (sterilized distilled water for control) containing AgNPs and incubated in the dark at 23 ± 2 °C. The growth of both the plumule and radical longer than 2 cm from their junctions was considered indicative of germination after 3 days of incubation. Germinated seeds were transferred to vials (five germinated seeds per vial) containing 30 mL of test solution. The shoot and root lengths were measured from the root/shoot junction after 4 days of incubation (25 °C, 70 rpm). The relative root length (RRL) and relative shoot length (RSL) were expressed as the percentage of inhibition (%) compared with the control.

## 5. Conclusions

Overall, this investigation demonstrated that biological methods have a different level of sensitivity to AgNPs, shown in the following toxic order; root growth > bacterial bioluminescence activity > algal growth > biosynthesis of enzymes > seed germination > enzymatic activity (Figure 4). In terms of particle sizes, the experimental results clearly confirm that high toxic effects with a small particle size were consistently observed for all activities compared to those of large ones: statistically significant differences for all bacterial activities were shown, while no statistical differences were shown for the activity of other types (algal growth, germination, and root growth) (Table 1 and Figure 4). Considering the total effects of all biological methods investigated (except shoot growth), it can be seen that the small AgNPs showed about 4.3-fold lower EC_50_ values (high toxicity) than the large AgNPs. Although the application of combining various biological methods may constitute a better tool for accurate assessment, the recommendable appropriate method on the basis of the discrimination between particle sizes and the sensitivity in the toxic assessment of AgNPs was found to be bacterial bioluminescence activity out of all the tested methods (Figure 4). Although the accurate contribution of the toxicity of particle sizes of AgNPs is unclear, the toxicity of AgNPs by metal ions might be insignificant and mostly mediated by the physical characteristics. However, the toxicity of NPs could be very variable under different laboratory and environmental conditions, and reactions with the constituents in environmental matrices, as well as the combined effect of the properties of NPs. Therefore, there is a need for a long-term study with real exposed conditions to understand the mechanisms involved in the toxicity of particle sizes of AgNPs for their safe and responsible usage.

## Figures and Tables

**Figure 1 ijms-21-08465-f001:**
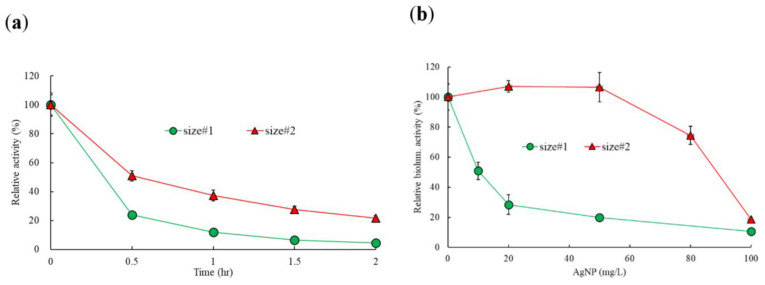
Response curves of the effects of two different sizes of AgNPs on bioluminescence activities of RB1436: (**a**) change of the percentage of relative bioluminescence activity during 2 h incubation periods with 50 mg/L AgNPs; (**b**) comparisons of the relative activity at the exposed dose ranges. Values are the mean ± SD (standard deviation) of triplicate samples.

**Figure 2 ijms-21-08465-f002:**
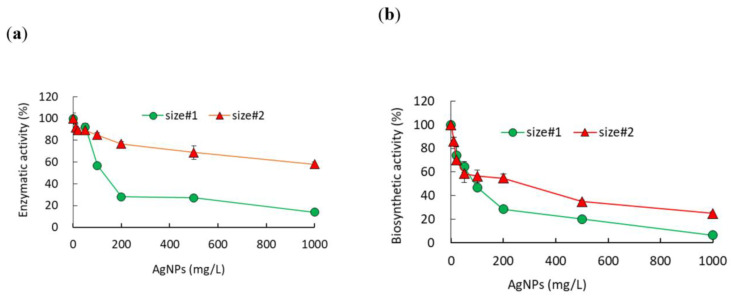
Comparisons of the effects of different sizes of AgNPs (**a**) on the enzymatic activity and (**b**) biosynthetic activity of the enzyme β-galactosidase.

**Figure 3 ijms-21-08465-f003:**
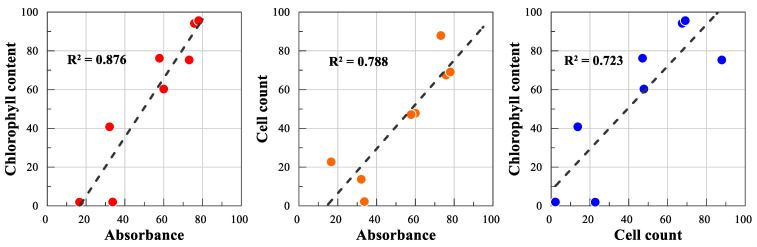
Correlations between the results of two endpoints measured to determine the effects of different sizes of AgNPs on the algal growth.

**Figure 4 ijms-21-08465-f004:**
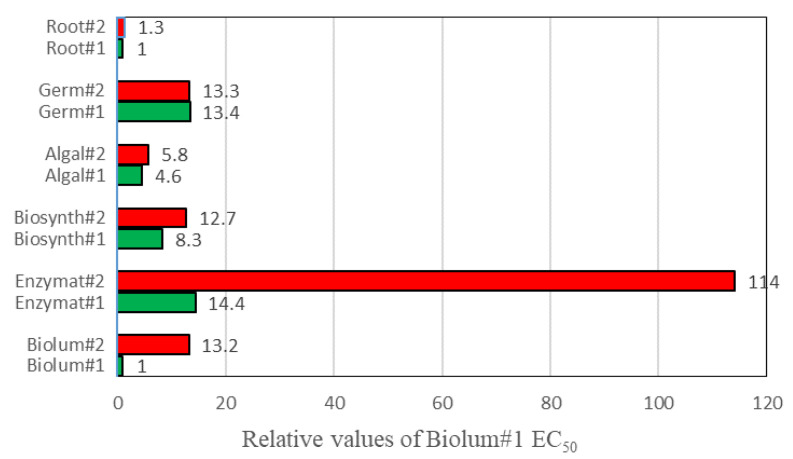
Comparison of the relative values of EC_50_ of the bacterial bioluminescence activity of particle size#1 (Biolum#1). A large value shows to a method with a low sensitivity, and a large value difference between two particles in the same method indicates a high level of discrimination between the tested particles. Label presents followings (e.g., Biolum#1 and Biolum#2 on the y-axis show the results of the exposure of particle size#1 and size#2 to the bacterial bioluminescence bioassay, respectively).

**Table 1 ijms-21-08465-t001:** Comparison of the effect of silver nanoparticles (AgNPs) to EC_50_ values calculated based on the various biological activities.

Bioassays		EC_50s_ for Each Particle Size (mg/L)	
		Size#1 (20–40 nm)	Size#2 (80–100 nm)
Bacterial bioluminescence		10.3 (7.72–13.75)	136 (124.4–147.6)
Enzymatic activity		148 (125–176)	1170 (930–1472)
Biosynthesis of enzyme		85 (70.2–99.8)	131 (110–152)
Algal growth		47.7(42.20–53.80)	60.1(46.01–79.97)
Plant	seed germination	138 (130.5–145.6)	137 (133.4–141.4)
	root growth	10 (7.4–13.5)	13.4 (8.9–20.1)
	shoot growth	94% activity at 200 mg/L	131% activity at 200 mg/L

**Table 2 ijms-21-08465-t002:** Comparisons of the effects of different sizes of AgNPs on the activity of algal growth, seed germination, and root and shoot growth.

Bioassays	Relative Activity (%) at							
	20 mg/L		50 mg/L		100 mg/L		200 mg/L	
	Size#1	Size#2	Size#1	Size#2	Size#1	Size#2	Size#1	Size#2
Algal growth	71 ± 13.8	86 ± 16.5	44 ± 7.1	40 ± 7.5	21 ± 8.0	21 ± 10.7	N.D.	19 ± 6.2
Germination	117 ± 21.5	113 ± 27.8	76 ± 26.0	106 ± 36.7	86 ± 36.3	91 ± 32.8	14 ± 15.8	3 ± 5.3
Root growth	28 ± 4.2	43 ± 16.1	27 ± 6.1	37 ± 7.9	28 ± 6.8	33 ± 4.8	26 ± 6.7	30 ± 3.8
Shoot growth	71 ± 14.0	97 ± 22.1	72 ± 12.0	100 ± 16.6	85 ± 13.2	89 ± 12.7	94 ± 14.0	97 ± 21.1

N.D. (not determined).

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
