# Peer review of "Evaluation of the Effects of Particle Sizes of Silver Nanoparticles on Various Biological Systems"

_ijms, 2020, doi:10.3390/ijms21228465_

Round 1
Reviewer 1 Report
What I really cannot understand is how the results are rationalized.
Authors study and show AgNP effects not only with different methods, but also on different microrganisms, so how can thay derive some general results?
I mean: if a small difference between the toxicity of smaller and biggere AgNP is observed, is it following from the method used or from the microorganism tested?
In other words, I cannot clearly understand which is the main interest in the study:
is to screen for toxicty on different microrganisms types?
or of different AgNP dimensions?
or the most appropriate method to assess toxicity?
Authors shoudl explain better the focus of the research, as in the paper it is not very clear, and not clearly presented.
Author Response
Answers for Reviewers’ Comments
Reviewer#1
Comments: What I really cannot understand is how the results are rationalized.
Authors study and show AgNP effects not only with different methods, but also on different microrganisms, so how can thay derive some general results?
I mean: if a small difference between the toxicity of smaller and biggere AgNP is observed, is it following from the method used or from the microorganism tested? In other words, I cannot clearly understand which is the main interest in the study: is to screen for toxicty on different microrganisms types? or of different AgNP dimensions? or the most appropriate method to assess toxicity?
Authors shoudl explain better the focus of the research, as in the paper it is not very clear, and not clearly presented.
Answer: Thank you for your valuable comments. In practice, the main objective of this study was to differentiate the effects of two different particle sizes of silver NPs on various biological systems. However, in the process of analyzing the results of this study, the bacterial activity, especially bioluminescence activity, showed very distinct results compared to other activities. On the basis of these outcomes, authors decided to propose the activity of bacterial bioluminescence as an appropriate bioassay for the evaluation of Ag-NPs among the methods investigated. Some ambiguity about the main purpose of this article was implicated in the process of mentioning these propose. To clarify the main purpose of this manuscript, following corrections were made in this revision:
Line 11-12 revised
Line 83-87 revised (purpose of this research)
Line 212-217 and 251-255: removed and rephrased of duplicated ones
Line 299~ : material and method section (a new reference [75] and information added)
All other corrections can be checked through the “track” function. In addition, the manuscript was checked again carefully and corrected in detail.
Reviewer 2 Report
Review of article ijms -990305
The manuscript is within the scope of the journal, well written and the results appear consistent. However, I have a couple of comments that the authors need to address:
(i) The methods (for example, the ICP-OES) can be elaborated. A good example of elaboration can be found in the following article: https://www.mdpi.com/1420-3049/24/1/98/htm
(ii) The discussions from 212 – 217 and from 251 – 255 are similar/duplicated. Please remove/rephrase one of them.
Author Response
Answers for Reviewers’ Comments
Reviewer#2
Comments: The manuscript is within the scope of the journal, well written and the results appear consistent. However, I have a couple of comments that the authors need to address:
(i) The methods (for example, the ICP-OES) can be elaborated. A good example of elaboration can be found in the following article: https://www.mdpi.com/1420-3049/24/1/98/htm
(ii) The discussions from 212 – 217 and from 251 – 255 are similar/duplicated. Please remove/rephrase one of them.
Answer: Thank you for your valuable comments. Authors revised carefully based on two reviewers’ comments. All corrections can be checked through the “track” function.
Line 212-217 and 251-255: removed and rephrased of duplicated ones
Line 299-302: materials and methods section: new information and reference [75] were added based on the comments.
All other corrections can be checked through the “track” function. In addition, the manuscript was checked carefully and corrected in detail.
Round 2
Reviewer 1 Report
paper is now a little bit more clear